# Overfitting or perfect fitting? Risk bounds for classification and regression rules that interpolate

**Mikhail Belkin**
The Ohio State University

**Daniel Hsu**
Columbia University

**Partha P. Mitra**
Cold Spring Harbor Laboratory

## Abstract

Many modern machine learning models are trained to achieve zero or near-zero training error in order to obtain near-optimal (but non-zero) test error. This phenomenon of strong generalization performance for "overfitted" / interpolated classifiers appears to be ubiquitous in high-dimensional data, having been observed in deep networks, kernel machines, boosting and random forests. Their performance is consistently robust even when the data contain large amounts of label noise.

Very little theory is available to explain these observations. The vast majority of theoretical analyses of generalization allows for interpolation only when there is little or no label noise. This paper takes a step toward a theoretical foundation for interpolated classifiers by analyzing local interpolating schemes, including geometric simplicial interpolation algorithm and singularly weighted $k$-nearest neighbor schemes. Consistency or near-consistency is proved for these schemes in classification and regression problems. Moreover, the nearest neighbor schemes exhibit optimal rates under some standard statistical assumptions.

Finally, this paper suggests a way to explain the phenomenon of adversarial examples, which are seemingly ubiquitous in modern machine learning, and also discusses some connections to kernel machines and random forests in the interpolated regime.

## 1 Introduction

The central problem of supervised inference is to predict labels of unseen data points from a set of labeled training data. The literature on this subject is vast, ranging from classical parametric and non-parametric statistics [48, 49] to more recent machine learning methods, such as kernel machines [39], boosting [36], random forests [15], and deep neural networks [25]. There is a wealth of theoretical analyses for these methods based on a spectrum of techniques including non-parametric estimation [46], capacity control such as VC-dimension or Rademacher complexity [40], and regularization theory [42]. In nearly all of these results, theoretical analysis of generalization requires "what you see is what you get" setup, where prediction performance on unseen test data is close to the performance on the training data, achieved by carefully managing the bias-variance trade-off. Furthermore, it is widely accepted in the literature that interpolation has poor statistical properties and should be dismissed out-of-hand. For example, in their book on non-parametric statistics, Györfi et al. [26, page 21] say that a certain procedure "may lead to a function which interpolates the data and hence is not a reasonable estimate".

Yet, this is not how many modern machine learning methods are used in practice. For instance, the best practice for training deep neural networks is to first perfectly fit the training data [35]. The resulting (zero training loss) neural networks after this first step can already have good performance on test data [53]. Similar observations about models that perfectly fit training data have been

---

E-mail: `mbelkin@cse.ohio-state.edu`, `djhsu@cs.columbia.edu`, `mitra@cshl.edu`

made for other machine learning methods, including boosting [37], random forests [19], and kernel machines [12]. These methods return good classifiers even when the training data have high levels of label noise [12, 51, 53].

An important effort to show that fitting the training data exactly can under certain conditions be theoretically justified is the margins theory for boosting [37] and other margin-based methods [6, 24, 28, 29, 34]. However, this theory lacks explanatory power for the performance of classifiers that perfectly fit *noisy* labels, when it is known that no margin is present in the data [12, 51]. Moreover, margins theory does not apply to regression and to functions (for regression or classification) that interpolate the data in the classical sense [12].

In this paper, we identify the challenge of providing a rigorous understanding of generalization in machine learning models that interpolate training data. We take first steps towards such a theory by proposing and analyzing interpolating methods for classification and regression with non-trivial risk and consistency guarantees.

**Related work.**  Many existing forms of generalization analyses face significant analytical and conceptual barriers to being able to explain the success of interpolating methods.

*Capacity control.*  Existing capacity-based bounds (e.g., VC dimension, fat-shattering dimension, Rademacher complexity) for empirical risk minimization [3, 4, 7, 28, 37] do not give useful risk bounds for functions with zero empirical risk whenever there is non-negligible label noise. This is because function classes rich enough to perfectly fit noisy training labels generally have capacity measures that grow quickly with the number of training data, at least with the existing notions of capacity [12]. Note that since the training risk is zero for the functions of interest, the generalization bound must bound their true risk, as it equals the *generalization gap* (difference between the true and empirical risk). Whether such capacity-based generalization bounds exist is open for debate.

*Stability.*  Generalization analyses based on algorithmic stability [8, 14] control the difference between the true risk and the training risk, assuming bounded sensitivity of an algorithm's output to small changes in training data. Like standard uses of capacity-based bounds, these approaches are not well-suited to settings when training risk is identically zero but true risk is non-zero.

*Regularization.*  Many analyses are available for regularization approaches to statistical inverse problems, ranging from Tikhonov regularization to early stopping [9, 16, 42, 52]. To obtain a risk bound, these analyses require the regularization parameter $\lambda$ (or some analogous quantity) to approach zero as the number of data $n$ tends to infinity. However, to get (the minimum norm) interpolation, we need $\lambda \to 0$ while $n$ is fixed, causing the bounds to diverge.

*Smoothing.*  There is an extensive literature on local prediction rules in non-parametric statistics [46, 49]. Nearly all of these analyses require local smoothing (to explicitly balance bias and variance) and thus do not apply to interpolation. (Two exceptions are discussed below.)

Recently, Wyner et al. [51] proposed a thought-provoking explanation for the performance of AdaBoost and random forests in the interpolation regime, based on ideas related to "self-averaging" and localization. However, a theoretical basis for these ideas is not developed in their work.

There are two important exceptions to the aforementioned discussion of non-parametric methods. First, the nearest neighbor rule (also called 1-nearest neighbor, in the context of the general family of $k$-nearest neighbor rules) is a well-known interpolating classification method, though it is not generally consistent for classification (and is not useful for regression when there is significant amount of label noise). Nevertheless, its asymptotic risk can be shown to be bounded above by twice the Bayes risk [18].[1] A second important (though perhaps less well-known) exception is the non-parametric smoothing method of Devroye et al. [21] based on a singular kernel called the Hilbert kernel (which is related to Shepard's method [41]). The resulting estimate of the regression function interpolates the training data, yet is proved to be consistent for classification and regression.

The analyses of the nearest neighbor rule and Hilbert kernel regression estimate are not based on bounding generalization gap, the difference between the true risk and the empirical risk. Rather, the true risk is analyzed directly by exploiting locality properties of the prediction rules. In particular, the

prediction at a point depends primarily or entirely on the values of the function at nearby points. This inductive bias favors functions where local information in a neighborhood can be aggregated to give an accurate representation of the underlying regression function.

**What we do.**   Our approach to understanding the generalization properties of interpolation methods is to understand and isolate the key properties of local classification, particularly the nearest neighbor rule. First, we construct and analyze an interpolating function based on multivariate triangulation and linear interpolation on each simplex (Section 3), which results in a geometrically intuitive and theoretically tractable prediction rule. Like nearest neighbor, this method is not statistically consistent, but, unlike nearest neighbor, its asymptotic risk approaches the Bayes risk as the dimension becomes large, even when the Bayes risk is far from zero—a kind of "blessing of dimensionality"[2]. Moreover, under an additional margin condition the difference between the Bayes risk and our classifier is exponentially small in the dimension.

A similar finding holds for regression, as the method is nearly consistent when the dimension is high.

Next, we propose a *weighted & interpolated nearest neighbor* (*wiNN*) scheme based on singular weight functions (Section 4). The resulting function is somewhat less natural than that obtained by simplicial interpolation, but like the Hilbert kernel regression estimate, the prediction rule is statistically consistent in any dimension. Interestingly, conditions on the weights to ensure consistency become less restrictive in higher dimension—another "blessing of dimensionality". Our analysis provides the first known non-asymptotic rates of convergence to the Bayes risk for an interpolated predictor, as well as tighter bounds under margin conditions for classification. In fact, the rate achieved by wiNN regression is statistically optimal under a standard minimax setting[3].

Our results also suggest an explanation for the phenomenon of adversarial examples [44], which are seemingly ubiquitous in modern machine learning. In Section 5, we argue that interpolation inevitably results in adversarial examples in the presence of any amount of label noise. When these schemes are consistent or nearly consistent, the set of adversarial examples (where the interpolating classifier disagrees with the Bayes optimal) has small measure but is asymptotically dense. Our analysis is consistent with the empirical observations that such examples are difficult to find by random sampling [22], but are easily discovered using targeted optimization procedures, such as Projected Gradient Descent [30].

Finally, we discuss the difference between *direct* and *inverse* interpolation schemes; and make some connections to kernel machines, and random forests in (Section 6).

Proofs of the main results, along with an informal discussion of some connections to graph-based semi-supervised learning, are given in the full version of the paper [11] on arXiv.[4]

## 2   Preliminaries

The goal of regression and classification is to construct a predictor $\hat{f}$ given labeled training data $(x_1, y_1), \ldots, (x_n, y_n) \in \mathbb{R}^d \times \mathbb{R}$, that performs well on unseen test data, which are typically assumed to be sampled from the same distribution as the training data. In this work, we focus on *interpolating methods* that construct predictors $\hat{f}$ satisfying $\hat{f}(x_i) = y_i$ for all $i = 1, \ldots, n$.

Algorithms that perfectly fit training data are not common in statistical and machine learning literature. The prominent exception is the *nearest neighbor rule*, which is among of the oldest and best-understood classification methods. Given a training set of labeled example, the nearest neighbor rule predicts the label of a new point $x$ to be the same as that of the nearest point to $x$ within the training set. Mathematically, the predicted label of $x \in \mathbb{R}^d$ is $y_i$, where $i \in \arg\min_{i'=1,\ldots,n} \|x - x_{i'}\|$. (Here, $\|\cdot\|$ always denotes the Euclidean norm.) As discussed above, the classification risk of the nearest neighbor rule is asymptotically bounded by *twice* the Bayes (optimal) risk [18]. The nearest neighbor

rule provides an important intuition that such classifiers can (and perhaps should) be constructed using local information in the feature space.

In this paper, we analyze two interpolating schemes, one based on triangulating and constructing the simplicial interpolant for the data, and another, based on weighted nearest neighbors with singular weight function.

## 2.1 Statistical model and notations

We assume $(X_1, Y_1), \ldots, (X_n, Y_n), (X, Y)$ are iid labeled examples from $\mathbb{R}^d \times [0, 1]$. Here, $((X_i, Y_i))_{i=1}^n$ are the iid training data, and $(X, Y)$ is an independent test example from the same distribution. Let $\mu$ denote the marginal distribution of $X$, with support denoted by $\mathrm{supp}(\mu)$; and let $\eta \colon \mathbb{R}^d \to \mathbb{R}$ denote the conditional mean of $Y$ given $X$, i.e., the function given by $\eta(x) := \mathbb{E}(Y \mid X = x)$. For (binary) classification, we assume the range of $Y$ is $\{0, 1\}$ (so $\eta(x) = \mathbb{P}(Y = 1 \mid X = x)$), and we let $f^* \colon \mathbb{R}^d \to \{0, 1\}$ denote the Bayes optimal classifier, which is defined by $f^*(x) := \mathbb{1}_{\{\eta(x) > 1/2\}}$. This classifier minimizes the risk $\mathcal{R}_{0/1}(f) := \mathbb{E}[\mathbb{1}_{\{f(X) \neq Y\}}] = \mathbb{P}(f(X) \neq Y)$ under zero-one loss, while the conditional mean function $\eta$ minimizes the risk $\mathcal{R}_{\mathrm{sq}}(g) := \mathbb{E}[(g(X) - Y)^2]$ under squared loss.

The goal of our analyses will be to establish excess risk bounds for empirical predictors ($\hat{f}$ and $\hat{\eta}$, based on training data) in terms of their agreement with $f^*$ for classification and with $\eta$ for regression. For classification, the expected risk can be bounded as $\mathbb{E}[\mathcal{R}_{0/1}(\hat{f})] \leq \mathcal{R}_{0/1}(f^*) + \mathbb{P}(\hat{f}(X) \neq f^*(X))$, while for regression, the expected mean squared error is precisely $\mathbb{E}[\mathcal{R}_{\mathrm{sq}}(\hat{\eta}(X))] = \mathcal{R}_{\mathrm{sq}}(\eta) + \mathbb{E}[(\hat{\eta}(X) - \eta(X)^2]$. Our analyses thus mostly focus on $\mathbb{P}(\hat{f}(X) \neq f^*(X))$ and $\mathbb{E}[(\hat{\eta}(X) - \eta(X))^2]$ (where the probability and expectations are with respect to both the training data and the test example).

## 2.2 Smoothness, margin, and regularity conditions

Below we list some standard conditions needed for further development.

$(A, \alpha)$-*smoothness (Hölder).* For all $x, x'$ in the support of $\mu$, $|\eta(x) - \eta(x')| \leq A \cdot \|x - x'\|^\alpha$.

$(B, \beta)$-*margin condition [31, 45].* For all $t \geq 0$, $\mu(\{x \in \mathbb{R}^d : |\eta(x) - 1/2| \leq t\}) \leq B \cdot t^\beta$.

$h$-*hard margin condition [32].* For all $x$ in the support of $\mu$, $|\eta(x) - 1/2| \geq h > 0$.

$(c_0, r_0)$-*regularity [5].* For all $0 < r \leq r_0$ and $x \in \mathrm{supp}(\mu)$, $\lambda(\mathrm{supp}(\mu) \cap \mathrm{B}(x, r)) \geq c_0 \lambda(\mathrm{B}(x, r))$, where $\lambda$ is the Lebesgue measure on $\mathbb{R}^d$, and $\mathrm{B}(c, r) := \{x \in \mathbb{R}^d : \|x - c\| \leq r\}$ denotes the ball of radius $r$ around $c$.

The regularity condition from Audibert and Tsybakov [5] is not very restrictive. For example, if $\mathrm{supp}(\mu) = \mathrm{B}(0, 1)$, then $c_0 \approx 1/2$ and $r_0 \geq 1$.

**Uniform distribution condition.** In what follows, we mostly assume uniform marginal distribution $\mu$ over a certain domain. This is done for the sake of simplicity and is not an essential condition. For example, in every statement the uniform measure can be substituted (with a potential change of constants) by an arbitrary measure with density bounded from below.

# 3 Interpolating scheme based on multivariate triangulation

In this section, we describe and analyze an interpolating scheme based on multivariate triangulation. Our main interest in this scheme is in its natural geometric properties and the risk bounds for regression and classification which compare favorably to those of the original nearest neighbor rule (despite the fact that neither is statistically consistent in general).

## 3.1 Definition and basic properties

We define an interpolating function $\hat{\eta} \colon \mathbb{R}^d \to \mathbb{R}$ based on training data $((x_i, y_i))_{i=1}^n$ from $\mathbb{R}^d \times \mathbb{R}$ and a (multivariate) triangulation scheme $T$. This function is *simplicial interpolation* [20, 27]. We assume without loss of generality that the (unlabeled) examples $x_1, \ldots, x_n$ span $\mathbb{R}^d$. The triangulation

scheme $T$ partitions the convex hull $\widehat{C} := \mathrm{conv}(x_1, \ldots, x_n)$ of the unlabeled examples into non-degenerate simplices[5] with vertices at the unlabeled examples; these simplices intersect only at $<d$-dimensional faces. Each $x \in \widehat{C}$ is contained in at least one of these simplices; let $U_T(x)$ denote the set of unlabeled examples $(x_{(1)}, \ldots, x_{(d+1)})$ that are vertices for a simplex containing $x$. Let $L_T(x)$ be the corresponding set of labeled examples $((x_{(1)}, y_{(1)}), \ldots, (x_{(d+1)}, y_{(d+1)}))$.[6] For any point $x \in \widehat{C}$, we define $\hat{\eta}(x)$ to be the unique *linear interpolation* of $L_T(x)$ at $x$ (defined below). For points $x \notin \widehat{C}$, we arbitrarily assert $U_T(x) = L_T(x) = \perp$, and define $\hat{\eta}(x) := 1/2$.

Recall that a *linear (affine) interpolation* of $(v_1, y_1), \ldots, (v_{d+1}, y_{d+1}) \in \mathbb{R}^d \times \mathbb{R}$ at a new point $x \in \mathbb{R}^d$ is given by the system of equations $\hat{\beta}_0 + x^\mathsf{T}\hat{\beta}$, where $(\hat{\beta}_0, \hat{\beta})$ are (unique) solutions to the system of equations $\hat{\beta}_0 + v_i^\mathsf{T}\hat{\beta} = y_i$ for $i = 1, \ldots, d+1$.

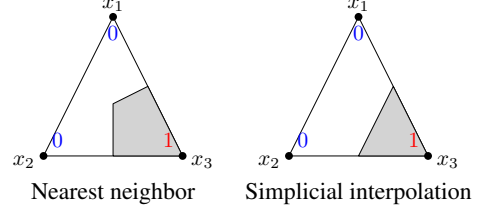

Figure 1: Comparison of nearest neighbor and simplicial interpolation. Consider three labeled examples from $\mathbb{R}^2 \times \{0, 1\}$: $(x_1, 0)$, $(x_2, 0)$, $(x_3, 1)$. Depicted in gray are the regions (within $\mathrm{conv}(x_1, x_2, x_3)$) on which the nearest neighbor classifier and simplicial interpolation classifier predict 1.

The predictions of the plug-in classifier based on simplicial interpolation are qualitatively very different from those of the nearest neighbor rule. This is true even when restricting attention to a single simplex. Suppose, for example, that $\eta(x) < 1/2$ for all $x \in \mathrm{conv}(x_1, \ldots, x_{d+1})$, so the Bayes classifier predicts 0 for all $x$ in the simplex. On the other hand, due to label noise, we may have some $y_i = 1$. Suppose in fact that only $y_{d+1} = 1$, while $y_i = 0$ for all $i = 1, \ldots, d$. In this scenario (depicted in Figure 1 for $d = 2$), the nearest neighbor rule (erroneously) predicts 1 on a larger fraction of the simplex than the plug-in classifier based on $\hat{\eta}$. The difference can be striking in high dimensions: $1/d$ for nearest neighbor versus $1/2^d$ for simplicial interpolation in $d$-dimensional version of Figure 1. This provides an intuition why, in contrast to the nearest neighbor rule, simplicial interpolation can yield to classifiers that are nearly optimal in high dimensions.

**Proposition 3.1.** *Suppose $v_1, \ldots, v_{d+1}$ are vertices of a non-degenerate simplex in $\mathbb{R}^d$, and $x$ is in their convex hull with barycentric coordinates $(w_1, \ldots, w_{d+1})$. The linear interpolation of $(v_1, y_1), \ldots, (v_{d+1}, y_{d+1}) \in \mathbb{R}^d \times \mathbb{R}$ at $x$ is given by $\sum_{i=1}^{d+1} w_i y_i$.*

One consequence of Proposition 3.1 for $\hat{\eta}$ is that if $x$ is contained in two adjacent simplices (that share a $<d$-dimensional face), then it does not matter which simplex is used to define $U_T(x)$; the value of $\hat{\eta}(x)$ is the same in any case. Geometrically, we see that the restriction of the interpolating linear function to a face of the simplex coincides with the interpolating linear function constructed on a sub-simplex formed by that face. Therefore, we deduce that $\hat{\eta}$ is a piecewise linear and continuous interpolation of the data $(x_1, y_1), \ldots, (x_n, y_n)$ on $\mathrm{conv}(x_1, \ldots, x_n)$.

We note that our prediction rule requires only locating the vertices of the simplex containing a given point, rather than the considerably harder problem of constructing a full triangulation. In fact, locating the containing simplex in a Delaunay triangulation reduces to solving polynomial-size linear programs [23]; in contrast, computing the full Delaunay triangulation has complexity exponential in the (intrinsic) dimension [2].

## 3.2 Mean squared error

We first illustrate the behavior of simplicial interpolation in a simple regression setting. Here, $(X_1, Y_1), \ldots, (X_n, Y_n), (X, Y)$ are iid labeled examples from $\mathbb{R}^d \times [0, 1]$. For simplicity, we assume that $\mu$ is the uniform distribution on a full-dimensional compact and convex subset of $\mathbb{R}^d$.

In general, each $Y_i$ may deviate from its conditional mean $\eta(X_i)$ by a non-negligible amount, and hence any function that interpolates the training data is "fitting noise". Nevertheless, in high dimension, the mean squared error of such a function will be quite close to that of the (optimal) conditional mean function.

**Theorem 3.2.** *Assume $\mu$ is the uniform distribution on a full-dimensional compact and convex subset of $\mathbb{R}^d$; $\eta$ satisfies the $(A, \alpha)$-smoothness condition; and the conditional variance function $x \mapsto \mathrm{var}(Y \mid X = x)$ satisfies the $(A', \alpha')$-smoothness condition. Let $\hat{\delta}_T := \sup_{x \in \widehat{C}} \mathrm{diam}(\mathrm{conv}(U_T(x)))$ denote the maximum diameter of any simplex in the triangulation $T$ derived from $X_1, \ldots, X_n$. Then*

$$\mathbb{E}[(\hat{\eta}(X) - \eta(X))^2] \leq \frac{1}{4}\mathbb{E}[\mu(\mathbb{R}^d \setminus \widehat{C})] + A^2\mathbb{E}[\hat{\delta}_T^{2\alpha}] + \frac{2}{d+2}A'\mathbb{E}[\hat{\delta}_T^{\alpha'}] + \frac{2}{d+2}\mathbb{E}[(Y - \eta(X))^2].$$

**Corollary 3.3.** *In addition to the assumptions in Theorem 3.2, assume $\mathrm{supp}(\mu)$ is a simple polytope in $\mathbb{R}^d$ and $T$ is constructed using Delaunay triangulation. Then $\limsup_{n \to \infty} \mathbb{E}[(\hat{\eta}(X) - \eta(X))^2] \leq \frac{2}{d+2}\mathbb{E}[(Y - \eta(X))^2].$*

### 3.3 Classification risk

We now analyze the statistical risk of the plug-in classifier based on $\hat{\eta}$, given by $\hat{f}(x) := \mathbb{1}_{\{\hat{\eta}(x) > 1/2\}}$. As in Section 3.2, we assume that $\mu$ is the uniform distribution on a full-dimensional compact and convex subset of $\mathbb{R}^d$.

We first observe that under the same conditions as Corollary 3.3, the asymptotic excess risk of $\hat{f}$ is $O(1/\sqrt{d})$. Moreover, when the conditional mean function satisfies a margin condition, this $1/\sqrt{d}$ can be replaced with a quantity that is exponentially small in $d$, as we show next.

**Theorem 3.4.** *Suppose $\eta$ satisfies the $h$-hard margin condition. As above, assume $\mu$ is the uniform distribution on a simple polytope in $\mathbb{R}^d$, and $T$ is constructed using Delaunay triangulation. Furthermore, assume $\eta$ is Lipschitz away from the class boundary (i.e., on $\{x \in \mathrm{supp}(\mu) : |\eta(x) - 1/2| > 0\}$) and that the class boundary $\partial$ has finite $d-1$-dimensional volume[7]. Then, for some absolute constants $c_1, c_2 > 0$ (which may depend on $h$), $\limsup_{n \to \infty} \mathbb{E}[\mathcal{R}_{0/1}(\hat{f})] \leq \mathcal{R}_{0/1}(f^*) \cdot (1 + c_1 e^{-c_2 d}).$*

*Remark* 3.5. The asymptotic risk bounds show that the risk of $\hat{f}$ can be very close to the Bayes risk in high dimensions, thus exhibiting a certain "blessing of dimensionality". This stands in contrast to the nearest neighbor rule, whose asymptotic risk does not diminish with the dimension and is bounded by twice the Bayes risk, $2\mathcal{R}_{0/1}(f^*)$.

## 4 Interpolating nearest neighbor schemes

In this section, we describe a weighted nearest neighbor scheme that, like the 1-nearest neighbor rule, interpolates the training data, but is similar to the classical (unweighted) $k$-nearest neighbor rule in terms of other properties, including convergence and consistency. (The classical $k$-nearest neighbor rule is not generally an interpolating method except when $k = 1$.)

### 4.1 Weighted & interpolated nearest neighbors

For a given $x \in \mathbb{R}^d$, let $x_{(i)}$ be the $i$-th nearest neighbor of $x$ among the training data $((x_i, y_i))_{i=1}^n$ from $\mathbb{R}^d \times \mathbb{R}$, and let $y_{(i)}$ be the corresponding label. Let $w(x, z)$ be a function $\mathbb{R}^d \times \mathbb{R}^d \to \mathbb{R}$. A weighted nearest neighbor scheme is simply a function of the form

$$\hat{\eta}(x) := \frac{\sum_{i=1}^k w(x, x_{(i)})y_{(i)}}{\sum_{i=1}^k w(x, x_{(i)})}.$$

In what follows, we investigate the properties of interpolating schemes of this type.

We will need two key observations for the analyses of these algorithms.

*Conditional independence.* The first key observation is that, under the usual iid sampling assumptions on the data, the first $k$ nearest neighbors of $x$ are conditionally independent given $X_{(k+1)}$. That implies that $\sum_{i=1}^k w(x, X_{(i)})Y_{(i)}$ is a sum of conditionally iid random variables[8]. Hence, under a mild condition on $w(x, X_{(i)})$, we expect them to concentrate around their expected value.

Assuming some smoothness of $\eta$, that value is closely related to $\eta(x) = \mathbb{E}(Y \mid X = x)$, thus allowing us to establish bounds and rates.

*Interpolation and singular weight functions.* The second key point is that $\hat{\eta}(x)$ is an interpolating scheme, provided that $w(x, z)$ has a singularity when $z = x$. Indeed, it is easily seen that if $\lim_{z \to x} w(x, z) = \infty$, then $\lim_{x \to x_i} \hat{\eta}(x) = y_i$. Extending $\hat{\eta}$ continuously to the data points yields a *weighted & interpolated nearest neighbor (wiNN)* scheme.

We restrict attention to singular weight functions of the following radial type. Fix a positive integer $k$ and a decreasing function $\phi \colon \mathbb{R}_+ \to \mathbb{R}_+$ with a singularity at zero, $\phi(0) = +\infty$. We take

$$w(x, z) := \phi\left( \frac{\|x - z\|}{\|x - x_{(k+1)}\|} \right).$$

Concretely, we will consider $\phi$ that diverge near $t = 0$ as $t \mapsto -\log(t)$ or $t \mapsto t^{-\delta}$, $\delta > 0$.

*Remark 4.1.* The denominator $\|x - x_{(k+1)}\|$ in the argument of $\phi$ is not strictly necessary, but it allows for convenient normalization in view of the conditional independence of $k$-nearest neighbors given $x_{(k+1)}$. Note that the weights depend on the sample and are thus data-adaptive.

*Remark 4.2.* Although $w(x, x_{(i)})$ are unbounded for singular weight functions, concentration only requires certain bounded moments. Geometrically, the volume of the region around the singularity needs to be small enough. For radial weight functions that we consider, this condition is more easily satisfied in high dimension. Indeed, the volume around the singularity becomes exponentially small in high dimension.

Our wiNN schemes are related to Nadaraya-Watson kernel regression [33, 50]. The use of singular kernels in the context of interpolation was originally proposed by Shepard [41]; they do not appear to be commonly used in machine learning and statistics, perhaps due to a view that interpolating schemes are unlikely to generalize or even be consistent; the non-adaptive Hilbert kernel regression estimate [21] (essentially, $k = n$ and $\delta = d$) is the only exception we know of.

## 4.2 Mean squared error

We first state a risk bound for wiNN schemes in a regression setting. Here, $(X_1, Y_1), \ldots, (X_n, Y_n), (X, Y)$ are iid labeled examples from $\mathbb{R}^d \times \mathbb{R}$.

**Theorem 4.3.** *Let $\hat{\eta}$ be a wiNN scheme with singular weight function $\phi$. Assume the following:*

1. *$\mu$ is the uniform distribution on a compact subset of $\mathbb{R}^d$ and satisfies the $(c_0, r_0)$ regularity condition for some $c_0 > 0$ and $r_0 > 0$.*

2. *$\eta$ satisfies the $(A, \alpha)$-smoothness for some $A > 0$ and $\alpha > 0$.*

3. *$\phi(t) = t^{-\delta}$ for some $0 < \delta < d/2$.*

*Let $Z_0 := \lambda(\mathrm{supp}(\mu))/\lambda(\mathrm{B}(0, 1))$, and assume $n > 2Z_0 k/(c_0 r_0^d)$. For any $x_0 \in \mathrm{supp}(\mu)$, let $r_{k+1,n}(x_0)$ be the distance from $x_0$ to its $(k+1)$st nearest neighbor among $X_1, \ldots, X_n$. Then*

$$\mathbb{E}\big[(\hat{\eta}(X) - \eta(X))^2\big] \leq A^2 \mathbb{E}[r_{k+1,n}(X)^{2\alpha}] + \bar{\sigma}^2\Big(k e^{-k/4} + \frac{d}{c_0(d - 2\delta)k}\Big),$$

*where $\bar{\sigma}^2 := \sup_{x \in \mathrm{supp}(\mu)} \mathbb{E}[(Y - \eta(x))^2 \mid X = x]$.*

The bound in Theorem 4.3 is stated in terms of the expected distance to the $(k+1)$st nearest neighbor raised to the $2\alpha$ power; this is typically bounded by $O((k/n)^{2\alpha/d})$. Choosing $k = n^{2\alpha/(2\alpha+d)}$ leads to a convergence rate of $n^{-2\alpha/(2\alpha+d)}$, which is minimax optimal.

## 4.3 Classification risk

We now analyze the statistical risk of the plug-in classifier $\hat{f}(x) = \mathbb{1}_{\{\hat{\eta}(x) > 1/2\}}$ based on $\hat{\eta}$.

As in Section 3.3, it is straigtforward obtain a risk bound for $\hat{f}$ under the same conditions as Theorem 4.3. Choosing $k = n^{2\alpha/(2\alpha+d)}$ leads to a convergence rate of $n^{-\alpha/(2\alpha+d)}$.

We now give a more direct analysis, largely based on that of Chaudhuri and Dasgupta [17] for the standard $k$-nearest neighbor rule, that leads to improved rates under favorable conditions. Our most general theorem along these lines is a bit lengthy to state, and hence we defer it to the full version of the paper. But a simple corollary is as follows.

**Corollary 4.4.** *Let $\hat{\eta}$ be a wiNN scheme with singular weight function $\phi$, and let $\hat{f}$ be the corresponding plug-in classifier. Assume the following:*

1. *$\mu$ is the uniform distribution on a compact subset of $\mathbb{R}^d$ and satisfies the $(c_0, r_0)$ regularity condition for some $c_0 > 0$ and $r_0 > 0$.*

2. *$\eta$ satisfies the $(A, \alpha)$-smoothness and $(B, \beta)$-margin conditions for some $A > 0$, $\alpha > 0$, $B > 0$, $\beta \geq 0$.*

3. *$\phi(t) = t^{-\delta}$ for some $0 < \delta < d/2$.*

*Let $Z_0 := \lambda(\mathrm{supp}(\mu))/\lambda(\mathrm{B}(0,1))$, and assume $k/n < p \leq c_0 r_0^d / Z_0$. Then for any $0 < \gamma < 1/2$,*

$$\mathbb{P}(\hat{f}(X) \neq f^*(X)) \leq B \left( \gamma + A \left( \frac{Z_0 p}{c_0} \right)^{\alpha/d} \right)^{\beta} + \exp \left( -\frac{np}{2} \left( 1 - \frac{k}{np} \right)^2 \right) + \frac{d}{4k\gamma^2 c_0 (d - 2\delta)}.$$

*Remark* 4.5. For consistency, we set $k := n^{(2+\beta)\alpha/((2+\beta)\alpha+d)}$, and in the bound, we plug-in $p := 2k/n$ and $\gamma := A(Z_0 p/c_0)^{\alpha/d}$. This leads to a convergence rate of $n^{-\alpha\beta/(\alpha(2+\beta)+d)}$.

*Remark* 4.6. The factor $1/k$ in the final term in Corollary 4.4 results from an application of Chebyshev inequality. Under additional moment conditions, which are satisfied for certain functions $\phi$ (e.g., $\phi(t) = -\log(t)$) with better-behaved singularity at zero than $t^{-\delta}$, it can be replaced by $e^{-\Omega(\gamma^2 k)}$. Additionally, while the condition $\phi(t) = t^{-\delta}$ is convenient for analysis, it is sufficient to assume that $\phi$ approaches infinity no faster than $t^{-\delta}$.

# 5 Ubiquity of adversarial examples in interpolated learning

The recently observed phenomenon of adversarial examples [44] in modern machine learning has drawn a significant degree of interest. It turns out that by introducing a small perturbation to the features of a correctly classified example (e.g., by changing an image in a visually imperceptible way or even by modifying a single pixel [43]) it is nearly always possible to induce neural networks to mis-classify a given input in a seemingly arbitrary and often bewildering way.

We will now discuss how our analyses, showing that Bayes optimality is compatible with interpolating the data, provide a possible mechanism for these adversarial examples to arise. Indeed, such examples are seemingly unavoidable in interpolated learning and, thus, in much of the modern practice. As we show below, any interpolating inferential procedure must have abundant adversarial examples in the presence of any amount of label noise. In particular, in consistent on nearly consistent schemes, like those considered in this paper, while the predictor agrees with the Bayes classifier on *the bulk* of the probability distribution, every "incorrectly labeled" training example (i.e., an example whose label is different from the output of the Bayes optimal classifier) has a small "basin of attraction" with every point in the basin misclassified by the predictor. The total probability mass of these "adversarial" basins is negligible given enough training data, so that a probability of misclassifying a randomly chosen point is low. However, assuming non-zero label noise, the union of these adversarial basins asymptotically is a dense subset of the support for the underlying probability measure and hence there are misclassified examples in every open set. This is indeed consistent with the extensive empirical evidence for neural networks. While their output is observed to be robust to random feature noise [22], adversarial examples turn out to be quite difficult to avoid and can be easily found by targeted optimization methods such as PCG [30]. We conjecture that it may be a general property or perhaps a weakness of interpolating methods, as some non-interpolating local classification rules can be robust against certain forms of adversarial examples [47].

To substantiate this discussion, we now provide a formal mathematical statement. For simplicity, let us consider a binary classification setting. Let $\mu$ be a probability distribution with non-zero density defined on a compact domain $\Omega \subset \mathbb{R}^d$ and assume non-zero label noise everywhere, i.e., for all $x \in \Omega$, $0 < \eta(x) < 1$, or equivalently, $\mathbb{P}(f^*(x) \neq Y \mid X = x) > 0$. Let $\hat{f}_n$ be a consistent interpolating classifier constructed from $n$ iid sampled data points (e.g., the classifier constructed in Section 4.3).

Let $\mathcal{A}_n = \{x \in \Omega : \hat{f}_n(x) \neq f^*(x)\}$ be the set of points at which $\hat{f}_n$ disagrees with the Bayes optimal classifier $f^*$; in other words, $\mathcal{A}_n$ is the set of "adversarial examples" for $\hat{f}_n$. Consistency of $\hat{f}$ implies that, with probability one, $\lim_{n \to \infty} \mu(\mathcal{A}_n) = 0$ or, equivalently, $\lim_{n \to \infty} \|\hat{f}_n - f^*\|_{L^2_\mu} = 0$. On the other hand, the following result shows that the sets $\mathcal{A}_n$ are asymptotically dense in $\Omega$, so that there is an adversarial example arbitrarily close to any $x$.

**Theorem 5.1.** *For any $\epsilon > 0$ and $\delta \in (0,1)$, there exists $N \in \mathbb{N}$, such that for all $n \geq N$, with probability $\geq \delta$, every point in $\Omega$ is within distance $2\epsilon$ of the set $\mathcal{A}_n$.*

*Proof sketch.* Let $(X_1, Y_1), \ldots, (X_n, Y_n)$ be the training data used to construct $\hat{f}_n$. Fix a finite $\epsilon$-cover of $\Omega$ with respect to the Euclidean distance. Since $\hat{f}_n$ is interpolating and $\eta$ is never zero nor one, for every $i$, there is a non-zero probability (over the outcome of the label $Y_i$) that $\hat{f}_n(X_i) = Y_i \neq f^*(X_i)$; in this case, the training point $X_i$ is an adversarial example for $\hat{f}_n$. By choosing $n = n(\mu, \epsilon, \delta)$ large enough, we can ensure that with probability at least $\delta$ over the random draw of the training data, every element of the cover is within distance $\epsilon$ of at least one adversarial example, upon which every point in $\Omega$ is within distance $2\epsilon$ (by triangle inequality) of the same. □

A similar argument for regression shows that while an interpolating $\hat{\eta}$ may converge to $\eta$ in $L^2_\mu$, it is generally impossible for it to converge in $L_\infty$ unless there is no label noise. An even more striking result is that for the Hilbert scheme of Devroye et al., the regression estimator almost surely *does not* converge at any fixed point, even for the simple case of a constant function corrupted by label noise [21]. This means that with increasing sample size $n$, at any given point $x$ misclassification will occur an infinite number of times with probability one. We expect similar behavior to hold for the interpolation schemes presented in this paper.

# 6 Discussion and connections

In this paper, we considered two types of algorithms, one based on simplicial interpolation and another based on interpolation by weighted nearest neighbor schemes. It may be useful to think of nearest neighbor schemes as *direct* methods, not requiring optimization, while our simplicial scheme is a simple example of an *inverse* method, using (local) matrix inversion to fit the data. Most popular machine learning methods, such as kernel machines, neural networks, and boosting, are inverse schemes. While nearest neighbor and Nadaraya-Watson methods often show adequate performance, they are rarely best-performing algorithms in practice. We conjecture that the simplicial interpolation scheme may provide insights into the properties of interpolating kernel machines and neural networks.

To provide some evidence for this line of thought, we show that in one dimension simplicial interpolation is indeed a special case of interpolating kernel machine. We will briefly sketch the argument without going into the details. Consider the space $\mathcal{H}$ of real-valued functions $f$ with the norm $\|f\|^2_{\mathcal{H}} = \int (\mathrm{d}f/\mathrm{d}x)^2 + \kappa^2 f^2 \, \mathrm{d}x$. This space is a reproducing kernel Hilbert Space corresponding to the Laplace kernel $e^{-\kappa|x-z|}$. It can be seen that as $\kappa \to 0$ the minimum norm interpolant $f^* = \arg\min_{f \in \mathcal{H}, \forall_i f(x_i) = y_i} \|f\|_{\mathcal{H}}$ is simply linear interpolation between adjacent points on the line. Note that this is the same as our simplicial interpolating method.

Interestingly, a version of random forests similar to PERT [19] also produces linear interpolation in one dimension (in the limit, when infinitely many trees are sampled). For simplicity assume that we have only two data points $x_1 < x_2$ with labels 0 and 1 respectively. A tree that correctly classifies those points is simply a function of the form $\mathbb{1}_{\{x > t\}}$, where $t \in [x_1, x_2]$. Choosing a random $t$ uniformly from $[x_1, x_2]$, we observe that $\mathbb{E}_{t \in [x_1, x_2]} \, \mathbb{1}_{\{x > t\}}$ is simply the linear function interpolating between the two data points. The extension of this argument to more than two data points in dimension one is straightforward. It would be interesting to investigate the properties of such methods in higher dimension. We note that it is unclear whether a random forest method of this type should be considered a direct or inverse method. While there is no explicit optimization involved, sampling is often used instead of optimization in methods like simulated annealing.

Finally, we note that while kernel machines (which can be viewed as two-layer neural networks) are much more theoretically tractable than general neural networks, none of the current theory applies in the interpolated regime in the presence of label noise [12]. We hope that simplicial interpolation can shed light on their properties and lead to better understanding of modern inferential methods.

## Acknowledgements

We would like to thank Raef Bassily, Luis Rademacher, Sasha Rakhlin, and Yusu Wang for conversations and valuable comments. We acknowledge funding from NSF. DH acknowledges support from NSF grants CCF-1740833 and DMR-1534910. PPM acknowledges support from the Crick-Clay Professorship (CSHL) and H N Mahabala Chair (IITM). This work grew out of discussions originating at the Simons Institute for the Theory of Computing in 2017, and we thank the Institute for the hospitality. PPM and MB thank ICTS (Bangalore) for their hospitality at the 2017 workshop on Statistical Physics Methods in Machine Learning.

## Footnotes

[1]More precisely, the expected risk of the nearest neighbor rule converges to $\mathbb{E}[2\eta(X)(1 - \eta(X))]$, where $\eta$ is the regression function; this quantity can be bounded above by $2R^*(1 - R^*)$, where $R^*$ is the Bayes risk.

[2]This does not remove the usual curse of dimensionality, which is similar to the standard analyses of $k$-NN and other non-parametric methods.

[3]An earlier version of this article paper contained a bound with a worse rate of convergence based on a loose analysis. The subsequent work [13] found that a different Nadaraya-Watson kernel regression estimate (with a singular kernel) could achieve the optimal convergence rate; this inspired us to seek a tighter analysis of our wiNN scheme.

[4]https://arxiv.org/abs/1806.05161

[5]We say a simplex in $\mathbb{R}^d$ is non-degenerate if it has non-zero $d$-dimensional Lebesgue measure.

[6]Of course, some points $x$ have more than one containing simplex; we will see that the ambiguity in defining $U_T(x)$ and $L_T(x)$ for such points is not important.

[7]I.e., $\lim_{\epsilon \to 0} \mu(\partial + \mathrm{B}(0, \epsilon)) = 0$, where "+" denotes the Minkowski sum, i.e., the $\epsilon$-neighborhood of $\partial$.

[8]Note that these variables are not independent in the ordering given by the distance to $x$, but a random permutation makes them independent.

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
