[Reviews · NeurIPS 2018]

Reviewer 1



On further reflection and seeing the responses, I have substantially increased my score. I think the takeaway "interpolating classifiers/regressors need not overfit" is quite important, even if the algorithms studied here are fairly different from the ones people are actually concerned about. I would suggest re-emphasizing that this is the main point of the paper in the introduction, and additionally toning down some of the discussion about a "blessing of dimensionality" as mentioned in your response / below. Original review is below. ======= This paper studies generalization bounds for certain classifiers which interpolate training labels. This is related to the recent "controversy" in learning theory, brought to prominence by [43] and continued in [9, 41], that practical deep learning models (and some kernel-based models) lie far outside the regime of performance explained by current learning theory, for example having extremely high Rademacher complexity, and yet perform well in practice. This paper gives bounds for two particular nearest-neighbor-like models, which interpolate the training data and yet are argued to generalize well in high dimensions. This is perhaps a step towards understanding the regime of interpolating classifiers. The 1-nearest-neighbor model, of course, is not consistent (having asymptotic risk up to twice the Bayes risk). ----- The first model studied is one based on simplicial interpolation, related to a nearest-neighbor model, studied for both regression and classification. The model is not argued to be practical, particularly in high-dimensional, but is studied as an analogue. The bounds shown, however, do not actually show consistency. The excess risk is bounded by 2 sigma^2 / (d + 2), with sigma^2 >= sup_x var(Y | X = x); this is argued to decrease in high dimensions, a "blessing of dimensionality," which of course it does if sigma is fixed and d increases. But keeping sigma fixed as the dimension increases is not necessarily particularly natural; for example, if we consider independent N(0, s^2) noise in each new dimension as we grow d, then sigma^2 would grow linearly with d, giving a constant excess risk. Also, it seems likely that even the terms in the bound of Theorem 3.2 which go to zero in the setting of Corollary 3.3 do so at a rate which gets exponentially worse in the dimension, so referring only to a so-called "blessing of dimensionality" in this case seems misleading. A similar case is true for classification: neither special case of Theorem 3.4 gives consistency. Corollary 3.5's results are somewhat hard to interpret in the interplay between the smoothness conditions and the dimension; using Remark 3.6, we see that the direct dependence on d is approximately (d + 1)^{- beta / 2}, but of course (A, alpha, B, beta) will be affected by d. Corollary 3.7 seems to show a simpler-to-interpret result, except that its conditions are actually impossible to satisfy! Holder continuity implies continuity (for alpha > 0), so eta must be continuous; but the condition of 3.7 implies that eta never hits 1/2 on the support of mu. Since mu is also assumed to be uniform on a simple polytope, i.e. supp(mu) is connected, Corollary 3.7 can only possibly apply to the case where eta > .5 on the whole distribution or eta < .5 on the whole distribution, in which case f^* is uniformly 0 or 1, and the problem is trivial. We could imagine replacing the condition of 3.7 with something like Pr( |eta(x) - 1/2| > h ) >= 1 - delta, giving an excess risk of delta + exp(-(d + 1) h^2 / 2). But even in this case, it seems likely that the relationship between h and delta would worsen with increasing dimension for reasonable functions eta. Thus, although there are regimes where high-dimensional simplicial interpolation becomes essentially-consistent, they are only very limited. Given that (a) the algorithm is utterly impractical in high dimensions, and (b) the sample complexity is likely also exponential in the dimension, it's unclear what this analysis really tells us about the cases that we actually care about. The connection to Laplace kernel methods and PERT-like trees in one dimension (in the discussion) is interesting, but it seems that the correspondence is lost in dimensions greater than one, while the rest of the paper focuses on the high-dimensional regime. It would be interesting to know if there is, for example, some other kernel than Laplace that gives a model like simplicial interpolation in higher dimensions. ----- The other model analyzed is for a weighted nearest neighbor classifier, with weights chosen such that the classifier interpolates the training labels. This classifier is perhaps more realistic to actually apply to data, and interestingly is also shown to be consistent in any dimension. The introduction (line ~93), however, claims that this technique "also benefits from the blessing of dimensionality as conditions on the weights to insure consistency become less restrictive in higher dimension." Though it is true that more weight functions are allowed as d increases, it seems like a stretch to call Corollary 4.4 an instance of "blessing of dimensionality": the convergence rate is exponential in d! (Also, as before, alpha and beta will likely increase with d for "reasonable" eta functions.) It would also be good to clarify (line 286) exactly what at least the rate for k is that provides the given rate; there doesn't seem to be a reason not to put at least the respective choices for gamma, p, and k in the appendix, to save the interested reader some time. ----- Overall: the problem setting of this paper is interesting, and the consistency of the weighted nearest neighbors classification scheme is intriguing (though it could use a little more comparison to the Hilbert kernel scheme). But its claims about a "blessing of dimensionality," and the overall value of the results about simplicial interpolation are misleading, and I'm not entirely sure what the greater takeaway of most of this work is really supposed to be. I did not carefully check the proofs. ----- Minor presentation notes and typos: - Line 28: "training training deep networks" - Line 91: "singular weigh functions" - Line 178: "can yield to classifiers" - should be "can yield" or "can lead to" - Theorem 4.3: The square in the denominator of kappa_p should be inside the expectation, not after the conditioning; it would also be clearer to add another set of brackets in the proof.

Reviewer 2



Summary The paper investigates two classification/regression rules that do not make any error on the training set, but for which certain guarantees about their generalization performance can still be established. While the first rule is only of theoretical interest illustrating possible effects, the second may also be of practical interest as it basically is a weighted nearest neighbor rule. Impression The results are certainly very interesting and as such deserve being accepted as they try to address an observation which has recently been made eg for neural networks: good generalization despite zero training error in the presence of noisy labels. However, the introduction is maybe a bit overselling, and the discussion on related work on page two is misleading since there mostly generalization bounds, that is, bounds that relate training to test error with high probability, are discussed, while the paper actually establishes bounds of a rather different flavor, see e.g. Theorem 3.2.

Reviewer 3



In this paper, the author(s) studies learning schemes based on interpolation. The main results include analysis for least squares regression, and classification and k-nearest neighbor schemes. It is well known that interpolation would lead to overfitting. The results in the paper hold because of the assumption that the marginal distribution is the uniform measure. This assumption is too restrictive: the data in many learning problems is irregular due to the large data dimension and data manifold structures. So strong analysis made in such a special setting should not be used to argue for the studied interpolated classification.